# THE GENERALIZATION-STABILITY TRADEOFF IN NEURAL NETWORK PRUNING

## ABSTRACT

Pruning neural network parameters is often viewed as a means to compress models, but pruning has also been motivated by the desire to prevent overfitting. This motivation is particularly relevant given the perhaps surprising observation that a wide variety of pruning approaches *increase* test accuracy despite sometimes massive reductions in parameter counts. To better understand this phenomenon, we analyze the behavior of pruning over the course of training, finding that pruning's effect on generalization relies more on the instability it generates (defined as the drops in test accuracy immediately following pruning) than on the final size of the pruned model. We demonstrate that even the pruning of unimportant parameters can lead to such instability, and show similarities between pruning and regularizing by injecting noise, suggesting a mechanism for pruning-based generalization improvements that is compatible with the strong generalization recently observed in over-parameterized networks.

## 1 INTRODUCTION

Pruning weights and/or convolutional filters from deep neural networks (DNNs) can substantially shrink parameter counts with minimal loss in accuracy (LeCun et al., 1990; Hassibi & Stork, 1993; Han et al., 2015a; Li et al., 2016; Molchanov et al., 2017; Louizos et al., 2017; Liu et al., 2017; Ye et al., 2018), enabling broader application of DNNs via reductions in memory-footprint and inference-FLOPs requirements. Moreover, many pruning methods have been found to actually improve generalization (measured by model accuracy on previously unobserved inputs) (Narang et al., 2017; Frankle & Carbin, 2018; You et al., 2019). Consistent with this, pruning was originally motivated as a means to prevent over-parameterized networks from overfitting to comparatively small datasets (LeCun et al., 1990).

Concern about over-parameterizing models has weakened, however, as many recent studies have found that *adding* parameters can actually *reduce* a DNN's generalization-gap (the drop in performance when moving from previously seen to previously unseen inputs), even though it has been shown that the same networks have enough parameters to fit large datasets of randomized data (Neyshabur et al., 2014; Zhang et al., 2016). Potential explanations for this unintuitive phenomenon have come via experiments (Keskar et al., 2016; Morcos et al., 2018; Yao et al., 2018; Belkin et al., 2018; Nagarajan & Kolter, 2019), and the derivation of bounds on DNN generalization-gaps that suggest less overfitting might occur as parameter counts increase (Neyshabur et al., 2018). This research has implications for neural network pruning, where a puzzling question has arisen: if larger parameter counts don't increase overfitting, how does pruning parameters throughout training improve generalization?

To address this question we first introduce the notion of pruning instability, which we define to be the size of the drop in network accuracy caused by a pruning iteration (Section 3). We then empirically analyze the instability and generalization associated with various magnitude-pruning (Han et al., 2015b) algorithms in different settings, making the following contributions:

1. We find a tradeoff between the stability and potential generalization benefits of pruning, and show iterative pruning's similarity to regularizing with noise—suggesting a mechanism unrelated to parameter counts through which pruning appears to affect generalization.

2. We characterize the properties of pruning algorithms which lead to instability and correspondingly higher generalization.

3. We derive a batch-normalized-parameter pruning algorithm to better control pruning stability.

## 2   RELATED WORK

There are various approaches to pruning neural networks. Pruning may be performed post-hoc (LeCun et al., 1990; Hassibi & Stork, 1993; Han et al., 2015b; Liu et al., 2017), or iteratively throughout training, such that there are multiple pruning events as the model trains (Hochreiter & Schmidhuber, 1997; Narang et al., 2017; Zhu & Gupta, 2017). Most methods prune parameters that appear unimportant to the function computed by the neural network, though means of identifying importance vary. Magnitude pruning (Han et al., 2015b) uses small-magnitude to indicate unimportance and has been shown to perform competitively with more sophisticated approaches (Gale et al., 2019).

Many pruning studies have shown that the pruned model has heightened generalization (Narang et al., 2017; Frankle & Carbin, 2018; You et al., 2019), consistent with the fact that pruning may be framed as a regularization (rather than compression) approach. For example, variational Bayesian approaches to pruning via sparsity-inducing priors (Molchanov et al., 2017; Louizos et al., 2017) can describe weight removal as a process that reduces model description length, which in theory may help improve generalization (Rissanen, 1978). Similarly, the idea that models may be described more succinctly at flat minima has motivated pruning in service of flat minimum search (Hochreiter & Schmidhuber, 1997). Though Dinh et al. (2017) notes, however, that flatness can be arbitrarily modified by reparameterizing the function, and sharp minima can generalize well.

VC dimension (a measure of model capacity) has motivated the use of iterative pruning to improve generalization (LeCun et al., 1990; Hassibi & Stork, 1993). Overfitting can be bounded above by an increasing function of VC dimension, which itself often increases with parameter counts, so fewer parameters can lead to a guarantee of less overfitting (Shalev-Shwartz & Ben-David, 2014). Unfortunately, such bounds can be so loose in practice that tightening them by reducing parameter counts need not translate to better generalization (Dziugaite & Roy, 2017).

Rather than support parameter-count-based arguments for generalization in DNNs, our results suggest iterative DNN pruning may improve generalization by creating various noisy versions of the internal representation of the data, which unpruned parameters try to fit to, as in noise-injection regularization (Srivastava et al., 2014; Poole et al., 2014). Dropout creates particularly similar noise, as it temporarily sets random subsets of layer outputs to zero (likely changing an input's internal representation every epoch). Indeed, applying dropout-like zeroing noise to a subset of features during training can encourage robustness to a post-hoc pruning of that subset (Leclerc et al., 2018; Gomez et al., 2018). Iterative DNN pruning noise ultimately differs, however, as it is: applied less frequently, not temporary (except in algorithms with weight re-entry), usually not random, and less well studied.

## 3   APPROACH

Given a neural network and set of test data, let $t$ be the top-1 test accuracy, the fraction of test data examples correctly classified multiplied by 100. We define a pruning algorithm's *instability* on pruning iteration $i$ in terms of $t$ measured immediately before ($t_{\mathrm{pre},i}$) and immediately after ($t_{\mathrm{post},i}$) pruning: $\mathrm{instability}_i = t_{\mathrm{pre},i} - t_{\mathrm{post},i}$. In other words, the instability is the size of the accuracy drop caused by a particular pruning event.

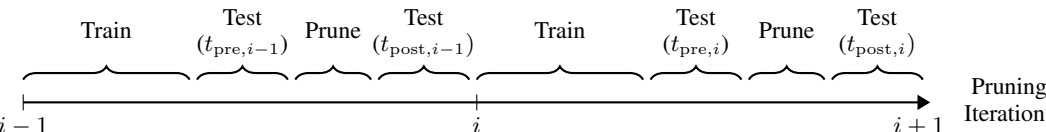

This measure is related to a weight's importance (sometimes referred to as "saliency"; LeCun et al. (1990); Hassibi & Stork (1993)) to the test accuracy, in that less stable pruning algorithms target more important sets of weights (all else equal). The stability of a pruning algorithm may be affected by many factors. Our experiments (Section 4) explore the effects of the following: pruning target, pruning schedule, iterative pruning rate, and model. The remainder of this section provides an overview of these factors and demonstrates a need for a novel pruning target, which we derive.

### 3.1 PRUNING TARGET

In all of our experiments, we use iterative magnitude pruning (Han et al., 2015b), which removes weights according to some magnitude-based rule, retrains the resulting smaller network to recover from the pruning, and repeats until the desired size reduction is met. We denote pruning algorithms that target the smallest-magnitude parameters with an "S" subscript (e.g. $\text{prune}_S$), random parameters with an "R" subscript, and the largest-magnitude parameters with an "L" subscript. The usual approach to pruning involves removing parameters that have the smallest magnitudes (Li et al., 2016; Gale et al., 2019), or, similarly, those parameters least important to the loss function as determined by some other metric (LeCun et al., 1990; Hassibi & Stork, 1993; Molchanov et al., 2016; 2017; Louizos et al., 2017; Ye et al., 2018; Yu et al., 2018; You et al., 2019).

#### 3.1.1 IDENTIFYING IMPORTANT BATCH-NORMALIZED PARAMETERS

The correlation between parameter magnitude and importance weakens in the presence of batch normalization (BN) (Ioffe & Szegedy, 2015). Without batch normalization, a convolutional filter with weights $W$ will produce feature map activations with half the magnitude of a filter with weights $2W$: filter magnitude clearly scales the output. With batch normalization, however, the feature maps are normalized to have zero mean and unit variance, and their ultimate magnitudes depend on the BN affine-transformation parameters $\gamma$ and $\beta$. As a result, in batch normalized networks, filter magnitude does not scale the output, and equating small magnitude and unimportance may therefore be particularly flawed. This has motivated approaches to use the scale parameter $\gamma$'s magnitude to find the convolutional filters that are important to the network's output (Ye et al., 2018; You et al., 2019). Here, we derive a novel approach to determining filter importance/magnitude that incorporates both $\gamma$ and $\beta$.

To approximate the expected value/magnitude of a batch-normalized, post-ReLU feature map activation, we start by defining the 2D feature map produced by convolution with BN:

$$M = \gamma \text{BN}(W * x) + \beta.$$

We approximate the activations within this feature map as $M_{ij} \sim \mathcal{N}(\beta, \gamma)$. This approximation is justified if central limit theorem assumptions are met by the dot products in $W * x$, and we empirically show in Figure A.1 that this approximation is highly accurate early in training, though it becomes less accurate as training progresses. Given this approximation, the post-ReLU feature map

$$R = \max\{0, M\}$$

has elements $R_{ij}$ that are either 0 or samples from a truncated normal distribution with left truncation point $l = 0$, right truncation point $r = \infty$, and mean $\mu$ where

$$\mu = \gamma \frac{\phi(\lambda) - \phi(\rho)}{Z} + \beta,$$

$$\lambda = \frac{l - \beta}{\gamma}, \rho = \frac{r - \beta}{\gamma}, Z = \Phi(\rho) - \Phi(\lambda),$$

and $\phi(x)$ and $\Phi(x)$ are the standard normal distribution's PDF and CDF (respectively) evaluated at $x$. Thus, an approximation to the expected value of $R_{ij}$ is given by

$$\mathbb{E}[R_{ij}] \approx \Phi(\lambda)0 + (1 - \Phi(\lambda))\mu.$$

We use the phrase "*E[BN] pruning*" to denote magnitude pruning that computes filter magnitude using this derived estimate of $\mathbb{E}[R_{ij}]$. E[BN] pruning has two advantages. First, this approach avoids the problematic assumption that filter importance is tied to filter $\ell_2$ norm in a batch-normalized network. Accordingly, we hypothesize that E[BN] pruning can grant better control of the stability of the neural network's output than pruning based on filters' $\ell_2$ norms. Second, the complexity of the calculation is negligible as it requires (per filter) just a handful of arithmetic operations on scalars, and two PDF and CDF evaluations, which makes it cheaper than a data-driven approach (e.g. approximating the expected value via the sample mean of feature map activations for a batch of feature maps).

## 3.2 SUMMARY OF MODELS

We consider three basic model classes: a simple network with convolutions (2x32, pool, 2x64, pool) and fully connected layers (512, 10) that we denote Conv4, VGG11 (Simonyan & Zisserman, 2014) with its fully-connected layers replaced by a single fully-connected layer, and ResNet18 (He et al., 2016). All convolutions are 3x3. We trained these models using Adam (Kingma & Ba, 2014) with initial learning rate $lr = 0.001$, as we found Adam more helpful than SGD for recovering from unstable pruning (seemingly consistent with the observation in Zhu & Gupta (2017) that recovery from pruning is more difficult when learning rates are low).

## 3.3 ITERATIVE PRUNING RATE AND SCHEDULE

For Conv4, we apply pruning to its first linear layer (which contains 94% of Conv4's 1,250,858 parameters). For VGG/ResNet, pruning targets the final four convolutional layers (which contain 90% of VGG11's 9,231,114 parameters, and 74% of ResNet18's 11,173,962 parameters). Pruning focused on later layers partly because, as also found in Li et al. (2016); You et al. (2019), it allowed the network to recover more easily.

The pruning algorithms we consider are iterative: we define a pruning schedule that describes the epochs on which pruning events occur, and set a corresponding (constant) iterative pruning rate that will ensure the total pruning percentage is met by the end of training (please see Appendix A.5 for rate and schedule details). Thus, throughout training, pruning steadily removes DNN parameters, with the iterative pruning rate determining the number pruned per event. While our plots label each pruning configuration with its *iterative* pruning rate, the *total* pruning percentages were: 42% of VGG11, 46% of ResNet18, and 10% of Conv4 (except in Appendix A.4, wherein we prune 85% of Conv4).

## 4 EXPERIMENTS

Pruning studies often aim to compress pre-trained models that generalize well, and consequently, much work has focused on metrics to identify parameter importance: if you can find the parameters that matter the least to the function computed by the DNN, then you can prune more parameters without significantly harming accuracy. As a bonus, such pruning methods can sometimes even *increase* generalization (Narang et al., 2017; Frankle & Carbin, 2018; You et al., 2019). However, the mechanism by which pruning induces higher generalization remains unclear. Here, rather than investigate how to best maintain accuracy when pruning the network, we instead focus on understanding the mechanisms underlying these generalization improvements.

## 4.1 THE GENERALIZATION-STABILITY TRADEOFF

Can improved generalization in pruned DNNs be explained by parameter-count reduction alone, or rather, do the properties of the pruning algorithm play an important role in generalization? As removing parameters from a DNN via pruning may make the DNN less capable of fitting to the noise in the training data, as originally suggested in LeCun et al. (1990); Hassibi & Stork (1993), we might expect that the generalization improvements observed in pruned DNNs are entirely explained by the number of parameters removed. In which case, methods that prune equal amounts of parameters would generalize similarly.

Alternatively, perhaps some aspect of the pruning algorithm itself is responsible for increased generalization. This seems plausible as the reported generalization benefits of pruning vary widely across studies. One possible explanation for this variability is differences in the pruning algorithms themselves. A key differentiator of these algorithms is their stability: more stable approaches may compute a very close approximation to the way the loss changes with respect to each parameter and prune a single parameter at a time (Hassibi & Stork, 1993), while less stable approaches may assume that parameter magnitude and importance are roughly similar and prune many weights all at once (Han et al., 2015b). Therefore, to the extent that differences in pruning algorithms explain differences in pruning-based generalization improvements, we might expect to observe a relationship between generalization and pruning stability.

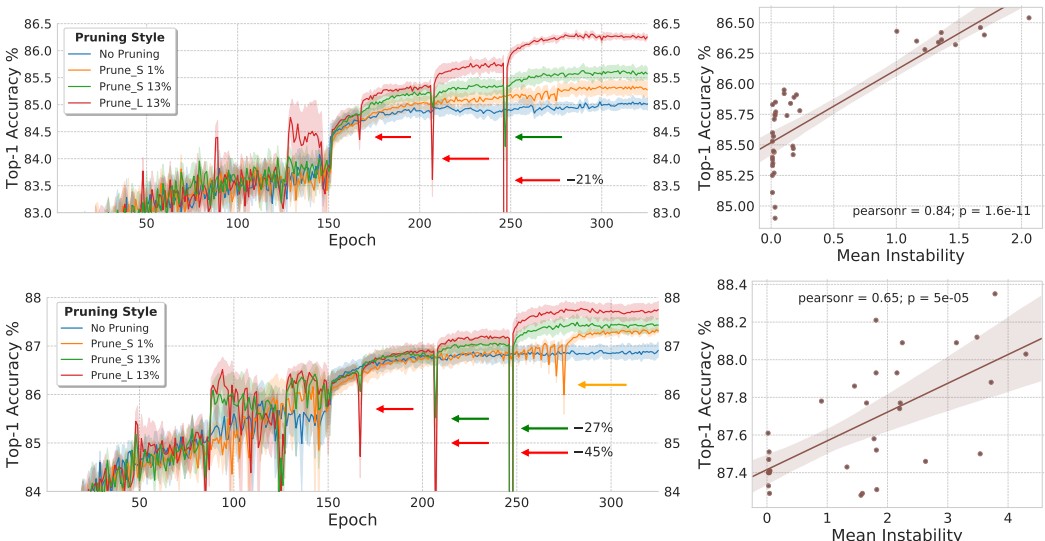

Figure 1: Pruning instability improves generalization of (Top) VGG11 and (Bottom) ResNet18 when training on CIFAR-10 (10 runs per configuration). (Left) Test accuracy during training of several models illustrates how adaptation to more unstable pruning leads to better generalization. (Right) Means reduce along the epoch dimension (creating one point per run-configuration combination).

To determine whether pruning algorithm stability affects generalization, we compared the instability and final top-1 test accuracy of several pruning algorithms with varying pruning targets and iterative pruning rates (Figure 1). Consistent with the nature of the pruning algorithm playing a role in generalization, we observed that more unstable pruning algorithms created *higher* final test accuracies than those which were stable (Figure 1, right; VGG11: Pearson's correlation $r = .84$, p-value $= 1.6e-11$; ResNet18: $r = .65$, p-value $= 5e-5$). While many pruning approaches have aimed to induce as little instability as possible, these results suggest that pruning techniques may actually facilitate better generalization when they induce *more* instability. Furthermore, these results suggest that parameter-count based arguments may not be sufficient to explain generalization in pruned DNNs, and suggest that the precise pruning method plays a critical role in this process.

Figure 1 also demonstrates that pruning events for prune$_L$ with a high iterative pruning rate (red curve, pruning as much as 13% of a given convolutional layer per pruning iteration) are substantially more destabilizing than other pruning events, but despite the dramatic pruning-induced drops in performance, the network recovers to higher performance within a few epochs. Several of these pruning events are highlighted with red arrows. Please see Appendix A.2 for visualization of the epoch-wise instabilities of each method in VGG11, and Appendix A.3 for an $\ell_2$-norm pruning version of Figure 1, which has qualitatively similar results.

Interestingly, we initially observed that ResNet18 adapted to pruning events more quickly than VGG11 (accuracy rebounded after pruning then flattened soon after instead of climbing steadily). Thinking that shortcut connections were allowing the network to adapt to pruning events too easily, we tried pruning a larger amount of the penultimate block's output layer: this reduced the number of shortcut connections to the final block's output layer, lengthened the adaptation period, and improved generalization. This simple improvement of pruning hyperparameters suggests a potential for further optimization of the results shown. Please see Appendix A.5.1 for all hyperparameters/details of these experiments.

## 4.2    THE ROLE OF WEIGHT MAGNITUDE IN PRUNING REGULARIZATION

We have demonstrated that, perhaps surprisingly, pruning larger magnitude weights via the E[BN] algorithm can result in *larger* test accuracy improvements (Figure 1). This suggests a positive correlation between pruning target magnitude and pruning's regularization effect. However, it's not clear whether this relationship holds more generally; i.e., perhaps it was caused by a feature of our

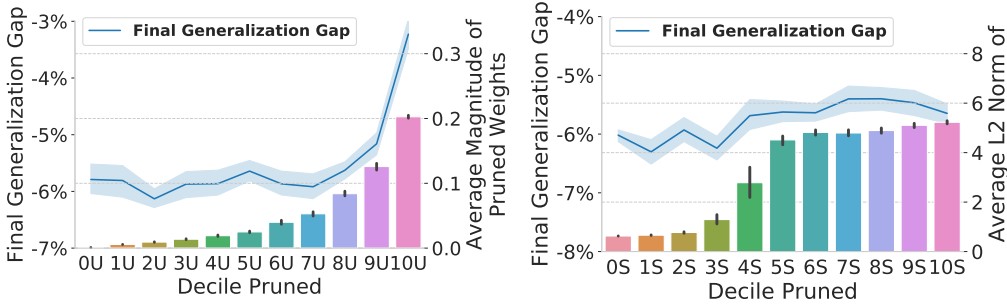

Figure 2: When pruning 10% of Conv4's largest dense layer, the final generalization gap depends on the magnitude of the weights that were pruned during training. This is particularly true when using unstructured pruning (left) rather than structured pruning (right).

E[BN] algorithm or the networks examined. Alternatively, this effect may be dependent on whether nodes/filters (structured pruning) or individual parameters (unstructured pruning) are pruned.

As such, we tested whether target weight magnitude correlates with pruning's regularizing effect when using both unstructured and structured magnitude pruning on the penultimate linear layer of a small network without batch normalization (Conv4). Specifically, we constructed a pruning target for each weight-magnitude decile (see Appendix A.5.2 for details), used each target to prune ten separate networks as they trained, and compared the generalization gaps (test-train accuracy) of the pruned networks to the target pruned (Figure 2).

For both unstructured and structured pruning (Figure 2 left and right, respectively), we found that pruning larger weights led to better generalization gaps, though, interestingly, this effect was much more dramatic in the context of unstructured pruning than structured pruning. One possible explanation for this is that, in structured pruning, the $\ell_2$ norm of pruned neurons did not vary dramatically past the fifth decile, whereas the unstructured deciles were approximately distributed exponentially. As a result, the top 50% of filters for the structured case were not clearly distinguished, making magnitude pruning much more susceptible to small sources of noise. These results suggest that, when weight magnitudes vary considerably, pruning large magnitude weights may lead to improved generalization.

Interestingly, for ResNet18, we actually found that *structured* prune$_L$ (red line) performed better than unstructured prune$_L$ (green line) (Figure 3). The worse performance of unstructured prune$_L$ may stem from its harming the helpful inductive bias provided by convolutional filters (i.e., perhaps removing the most important connections in all convolutional filters degrades performance more than pruning the same number of connections via removal of several entire filters) or its lower instability.

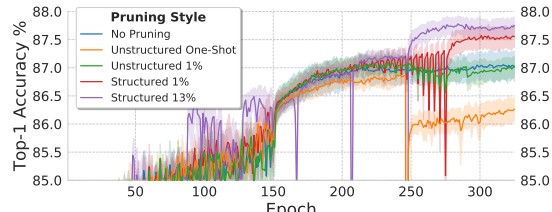

Figure 3: The top-1 test accuracy during training with multiple approaches to pruning ResNet18.

### 4.3 THE ROLE OF ITERATIVE PRUNING RATE IN PRUNING INSTABILITY

While pruning large magnitude weights appears to play a role in pruning's ability to improve generalization, more commonly used pruning algorithms often see generalization improvements when targeting the smallest magnitude or least important parameters, suggesting that target magnitude/importance is not the only characteristic of pruning algorithms relevant to generalization. One possibility is that, given a pruning target, pruning more parameters per pruning iteration (while holding constant the total pruning percentage) may lead to greater instability. If this is the case, the generalization-stability tradeoff suggests that the increase in instability from raising the iterative pruning rate would coincide with improved generalization performance. Alternatively, if the pruning target or total pruning percentage is all that matters, we may expect that changing the iterative

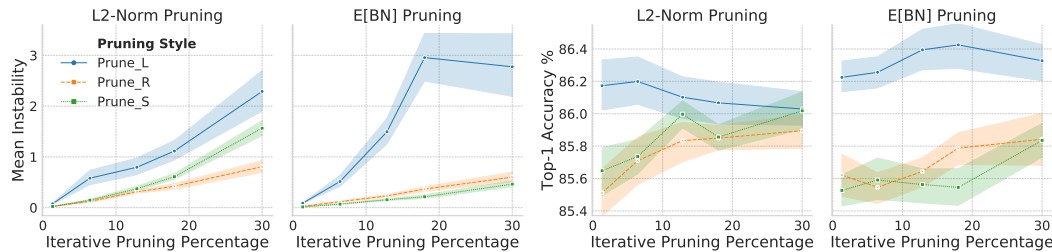

Figure 4: In VGG11, increasing the iterative pruning rate (and decreasing the number of pruning events in order to hold total pruning percentage constant) leads to more instability, and can allow methods that target less important parameters to generalize better. Additionally, E[BN] magnitude better approximates parameter importance than $\ell_2$-norm magnitude (see Figure A2 for another example and discussion of this phenomenon). An unpruned baseline model has 85.21% accuracy.

pruning rate (while keeping the pruning target and total pruning percentage fixed) would not affect generalization.

To test this, we plotted mean instability and test accuracy as a function of different iterative pruning rates for both $\ell_2$-norm and E[BN] pruning (Figure 4). Consistent with iterative pruning rate playing a role in instability, we find that (given a pruning target) more instability is induced by using larger iterative pruning rates (Figure 4 left). Moreover, pruning random or small magnitude parameters performs best at the largest iterative rate (30%), supporting the idea that these methods require a source of instability to boost generalization. Note this suggests that, when targeting less important weights, higher iterative pruning rates during training can be an effective way to induce additional instability and generalization. (Algorithm and experiment details are available in Appendix A.5.4.)

Perhaps strangely, higher iterative pruning rates did not translate to improved generalization when targeting the largest magnitude weights (prune$_L$) with $\ell_2$-norm pruning. The fact that prune$_L$ does not generalize the best at the highest iterative pruning rate may be due to the reduction in pruning iterations required by the large iterative pruning rate (i.e., when the iterative rate is at 30%, the number of pruning events is capped at three, which removes 90% of a layer). Thus, while this rate grants more instability (Figure 4 left) per iteration, pruning affects the network less often. The idea that the regularizing effect of pruning is enhanced by pruning more often may also help explain the observation that methods that prune iteratively can generalize better (Han et al., 2015b).

Another possibility is that, since raising the iterative pruning rate (and consequently the duration between pruning events) tends to make the $\ell_2$-norm worse for differentiating parameters by their importance to accuracy[1], raising the iterative pruning rate causes prune$_L$ with $\ell_2$-norm pruning to target less important weights. Consequently, prune$_L$ with $\ell_2$-norm pruning may generalize worse at higher iterative rates by leaving unpruned more important weights, the presence of which can harm model generalization (Hinton & Van Camp, 1993; Morcos et al., 2018). Relatedly, this also means that prune$_S$ with $\ell_2$-norm pruning may *increase* (in networks with batch normalization at least) instability and generalization by failing to avoid the pruning of important parameters.

## 4.4 ITERATIVE PRUNING AS NOISE INJECTION

Our results thus far suggest that pruning improves generalization when it creates instability throughout training. These prior results, though, involved damaging model capacity simply by the nature of pruning, which decreases the number of model parameters. It therefore remains possible that the generalization benefits we've seen rely upon the reduction in capacity conferred by pruning. Here, we examine this critical question.

We first note that iterative pruning can be viewed as noise injection (Srivastava et al., 2014; Poole et al., 2014), with the peculiarity that the noise permanently zeroes a subset of weights. Removing

---

[1]In Figure 4 left: with E[BN] pruning, prune$_L$ tends to become more unstable relative to prune$_S$ as the iterative rate increases, but with $\ell_2$-norm pruning, prune$_L$ and prune$_S$ create similar instabilities for all rates.

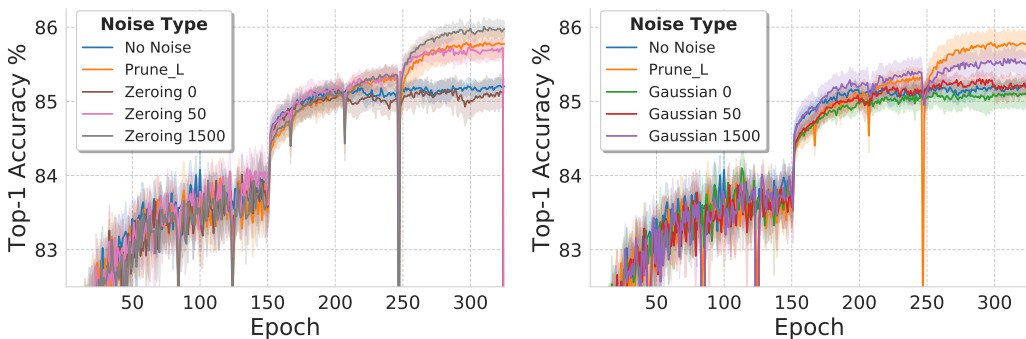

Figure 5: Generalization improvements from pruning bear resemblance to those obtained by using temporary (Left) multiplicative zeroing noise, and (Right) additive Gaussian noise, as long as the noise is applied for enough batches/steps.

the permanence of this zeroing can mitigate some of the capacity effect of pruning[2] and, therefore, help us isolate and study how iterative pruning regularizes through noise injection.

As a baseline, we consider $prune_L$ applied to VGG11, judging filter magnitude via the $\ell_2$-norm (additional experimental details are in Appendix A.5.5). We then modify this algorithm such that, rather than permanently prune filters, it simply multiplies the filter weights by zero, then allows the zeroed weights to immediately resume training in the network ("Zeroing 0" in Figure 5 Left). However, by allowing pruned weights to immediately recover, this experiment also removes a key, potentially regularizing aspect of pruning noise: the requirement that the rest of the network adapts to fit the new representations generated by pruning. To encourage this potentially important facet of pruning noise, we also added variants that held weights to zero for 50 and 1500 consecutive batches[3]. As a related experiment, we also measured the impact of adding Gaussian noise to the weights in Figure 5, right. Noise was applied either once (Gaussian 0) or repeatedly over a series of training batches (Gaussian 50/1500).

If the capacity effects of weight removal are not necessary to explain pruning's effect on generalization, then we would expect that the generalization behavior of these non-permanent noise injection algorithms could mimic the generalization behavior of $prune_L$. Alternatively, if weight removal is a necessary component of pruning-based generalization improvements, then we would not expect close similarities between the generalization phenomena of $prune_L$ and non-permanent pruning noise injection.

Consistent with the capacity effects of weight removal not being necessary to explain generalization in pruned DNNs, applying zeroing noise for 50 batches to filters (rather than pruning them completely) generates strikingly similar accuracy to $prune_L$ (Figure 5 Left). Specifically, the patterns in instability are qualitatively and quantitatively similar, as are the generalization levels throughout training.

Importantly, we found that applying zeroing noise once (Zeroing 0; brown line) was not sufficient to generate better performance, suggesting that the regularization induced by forcing weights to adapt to noised representations is critical to pruning's ability to improve generalization. Moreover, we found that, while applying Gaussian noise could increase generalization if applied for long enough (Gaussian 1500; purple line), it still did not match the performance of $prune_L$, suggesting that multiplicative zeroing noise is substantially more effective than additive Gaussian noise[4]. Together, these results demonstrate that pruning induced generalization benefits are not merely explained by weight removal, but rather are dependent on the regularization conferred by forcing networks to adapt to noised representations over a sufficiently long period throughout training.

---

[2]This approach still effectively removes any weights that do not learn after reentering the model. However, we observed (see Figure 5) that pruning the reentered weights at convergence resulted in a marked drop in performance (for all noise schemes except "Zeroing 1500"), showing that the reentered weights had typically learned after reentry, and that temporary zeroing is therefore less harmful to capacity than permanent pruning.

[3]The pruning noise we apply affects a particular weight for no more than one segment of training time (where a segment is a series of N consecutive batches)

[4]At the scale we used.

## 5 Discussion

In this study, we defined the notion of pruning algorithm instability, and applied several pruning approaches[5] to multiple neural networks, assessing the approaches' effects on instability and generalization. Throughout these experiments, we observed that pruning algorithms that generated more instability led to better generalization (as measured by test accuracy). For a given pruning target and total pruning percentage, instability and generalization could be fueled by raising iterative pruning rates (Figure 4, Section 4.3). Additionally, targeting more important weights, again holding total parameters pruned constant, led to more instability and generalization than targeting less important weights (Figure 1, Section 4.1).

These results support the idea that the generalization benefits of pruning cannot be explained solely by pruning's effect on parameter counts—the properties of the pruning algorithm must be taken into account. Our analysis also suggests that the capacity effects of weight-removal may not even be *necessary* to explain how pruning improves generalization. Indeed, we provide an interpretation of iterative pruning as noise injection, a popular approach to regularizing DNNs, and find that making pruning noise impermanent provides pruning-like generalization benefits while not removing as much capacity as permanent pruning (Figure 5, Section 4.4).

### 5.1 Caveats and Future Work

While not emphasized in our discussion, pruning algorithm *stability* can be a desirable property, as recovery from pruning damage is not guaranteed. Indeed, pruning too many large/important weights can lead to worse final generalization (Li et al., 2016). Recovery appears to be a function of several factors, including: learning rate (Zhu & Gupta, 2017)); presence of an ongoing regularization effect (Figure 3, Section 4.2); preservation of helpful inductive biases (Figure 3, Section 4.2); and damage to network capacity (e.g., removing too much of an important layer could cause underfitting).

A better understanding of the factors which aid recovery from pruning instability could aid the design of novel pruning algorithms. For example, pruning methods that allow weights to re-enter the network (Narang et al., 2017) could perhaps prune *important* weights occasionally to enhance generalization improvements, without risking permanent damage to the pruned networks (see Appendix A.4).

In describing how pruning regularizes a model, we touched on similarities between pruning and noise injection. Our results, however, may also be consistent with other parameter-count-independent approaches to understanding generalization in neural networks, as pruning may reduce the information stored in the network's weights (Hinton & Van Camp, 1993), and make the network more distributed (Morcos et al., 2018; Dettmers & Zettlemoyer, 2019). This raises the possibility that pruning noise engenders helpful properties in DNNs, though it remains unclear whether such properties might be identical to those achieved with more common noise injection schemes (Srivastava et al., 2014; Poole et al., 2014). Further exploration will be necessary to better understand the relationship between these approaches.

One important caveat of our results is that they were generated with CIFAR-10, a relatively small dataset, so future work will be required to evaluate whether the presented phenomena hold in larger datasets. Relatedly, we only studied pruning's regularizing effect in isolation and did not include commonly used regularizers (e.g., weight decay) in our setups. In future work, it would be interesting to examine whether pruning complements the generalization improvements of other commonly used regularization techniques.

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

# A    APPENDIX

## A.1    QUALITY OF NORMALITY APPROXIMATION BY LAYER AND TRAINING TIME

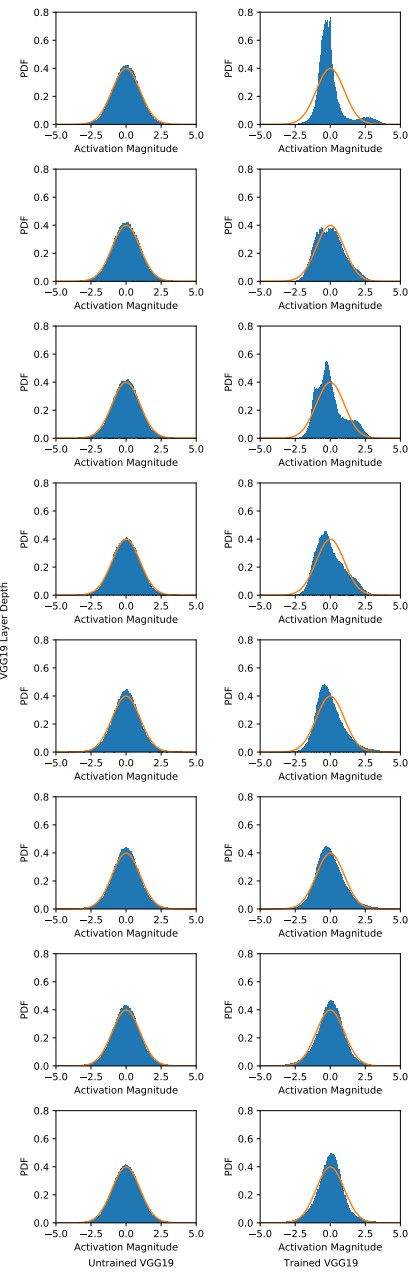

Figure A1: We examined the normalized activations (shown in blue histograms) of feature maps in the final eight convolutional layers of VGG19 before (left) and after (right) training to convergence. We found that the approximation to standard normality (shown in orange) of these activations is reasonable early on but degrades with training (particularly in layers near the output).

The main drawback to the E[BN] approach (Section 3.1.1) is the sometimes poor approximation $M_{ij} \sim N(\beta, \gamma)$. In Figure A.1, the approximation's quality depends on layer and training epoch. A less serious drawback is that this approach does not account for the strength of connections to the post-BN feature map, which could have activations with a large expected value but low importance if relatively small-magnitude weights connected the map to the following layer.

## A.2 EPOCH-WISE INSTABILITIES

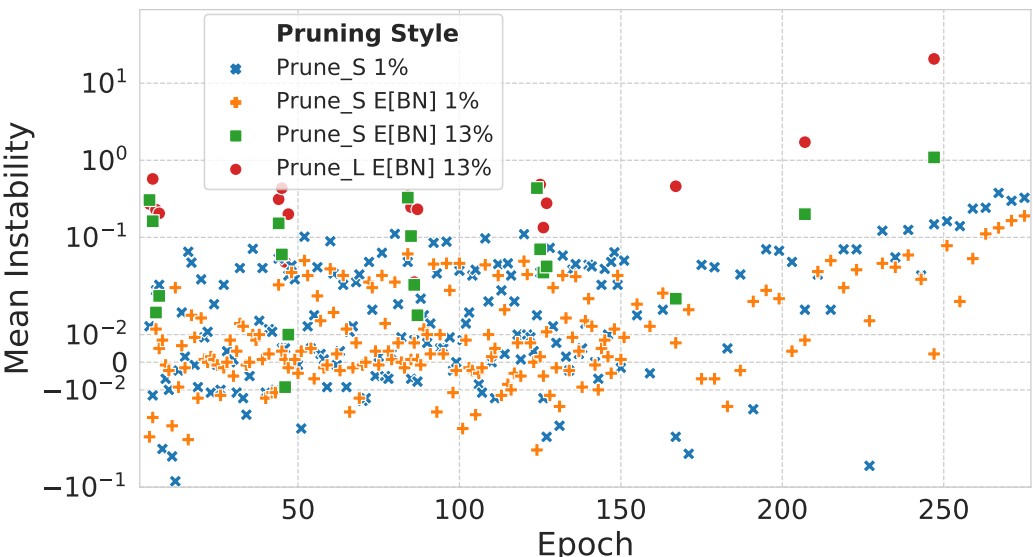

Figure A2: In VGG11, prune$_S$ E[BN] is more stable than prune$_S$, which uses filter-$\ell_2$-norm to compare parameter magnitudes. Methods with higher iterative pruning rates create more instability on a given iteration. Means reduce along the run dimension (10 runs per configuration).

Note that this graph uses a method (prune$_S$) that was included in Figure 1 right, but was not displayed in Figure 1 left due to its similarity to prune$_S$ E[BN]. Additional experimental details are in Section A.5.1.

### A.3 $\ell_2$-NORM-PRUNING VERSION OF FIGURE 1

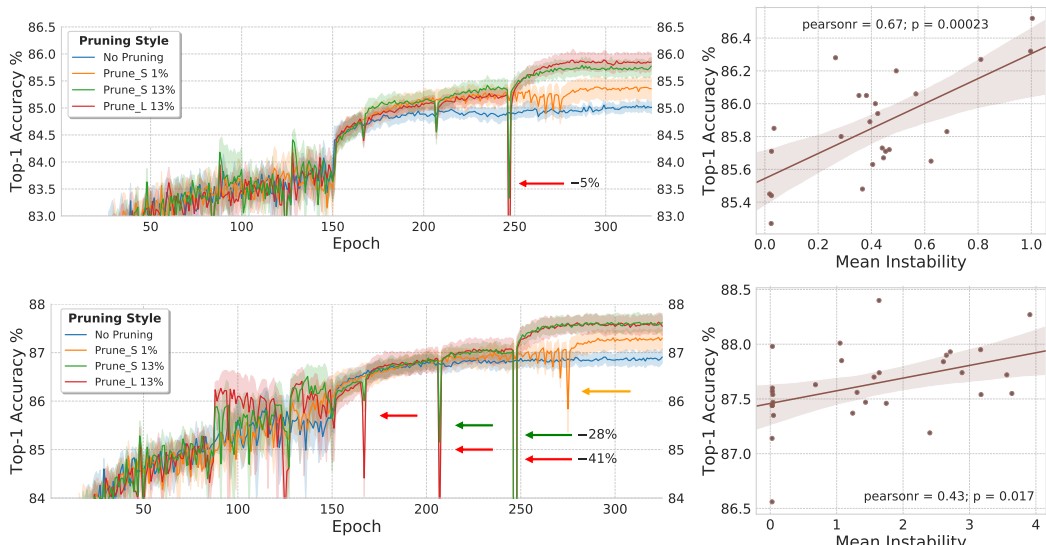

Figure A3: Pruning instability improves generalization of (Top) VGG11 and (Bottom) ResNet18 when training on CIFAR-10 (10 runs per configuration). (Left) Test accuracy during training of several models illustrates how adaptation to more unstable pruning leads to better generalization. (Right) Means reduce along the epoch dimension (creating one point per run-configuration combination).

Here we use the same training and pruning configurations that were used for Figure 1, but we replace E[BN] pruning with $\ell_2$-norm pruning. Qualitatively, the two figures' results are similar. Interestingly, though, the correlation between instability and generalization is somewhat weaker with $\ell_2$-norm pruning. This may be explained by the fact that $\ell_2$-norm pruning generates a narrower spectrum of instabilities, which is perhaps due to $\ell_2$-norm scoring's inability to accurately assess parameter importance (illustrated in Figure 4).

A.4    THE GENERALIZATION-STABILITY TRADEOFF AT 85% SPARSITY IN CONV4

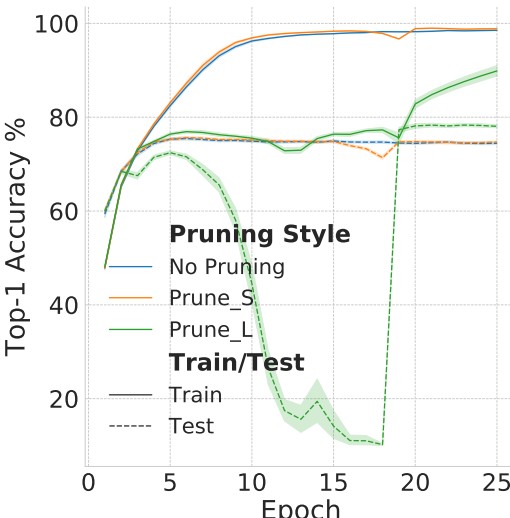

Figure A4: When training Conv4 on CIFAR10, unstable pruning can significantly improve the baseline's generalization. The training accuracies and test accuracies (the latter were calculated immediately after pruning) illustrate how much each pruning algorithm disturbs the neural network's output during training.

While it's unclear how much unstable pruning, which is particularly damaging to capacity, can be sustained at high sparsity levels, $prune_L$ can lead to generalization several percentage points above the baseline/$prune_S$ when pruning 85% of Conv4. Please see Section A.5.6 for experimental setup details.

## A.5 Experimental Details

Unstructured magnitude pruning entails removing individual weights (subsets of filters/neurons), which are selected for pruning based on their magnitude. Our unstructured pruning approach does not allow previously pruned weights to reenter the network (Narang et al., 2017; Zhu & Gupta, 2017; Gale et al., 2019). Structured magnitude pruning removes entire filters/neurons, which are selected based on their $\ell_2$-norms or via the E[BN] calculation. Except where noted, we use structured pruning for VGG11 and ResNet18.

We denote the pruning of $n$ layers of a network by specifying a series of epochs at which pruning starts $s = (s_1, ..., s_n)$, a series of epochs at which pruning ends $e = (e_1, ..., e_n)$, a series of fractions of parameters to remove $p = (p_1, ..., p_n)$, and an inter-pruning-iteration retrain period $r \in \mathbb{N}$. For a given layer $l$, the retrain period $r$ and fraction $p_l$ jointly determine the iterative pruning percentage $i_l$. Our experiments prune the same number of parameters $i_l \times \text{size}(layer_l)$ per pruning iteration, ultimately removing $p_l \times 100\%$ of the parameters by the end of epoch $e_l$.

Our approach is designed to study the effects of changing factors such as the iterative pruning rate and lacks some practically helpful features, e.g. hyperparameters indicating how many parameters can be safely pruned (Liu et al., 2017; Molchanov et al., 2017). When layerwise iterative pruning percentages differ (i.e., when there exists an $a$ and $b$ such that $i_a$ and $i_b$ are unequal), our figures state the largest iterative pruning rate that was used in any of the layers.

For ResNet, our pruning algorithms did not account for the magnitude of incoming shortcut connections when judging filter magnitude/importance. Though we did prune the incoming and outgoing shortcut connections associated with any pruned feature maps.

We used only the CIFAR-10 dataset (Krizhevsky & Hinton, 2009) in our experiments, a limitation of our study. We used batch size 128, and only used data augmentation in the decile experiment (Figure 2). For some experiments, we give multi-step learning rate schedules $lr_s = (x, y)$, which means we shrink the learning rate by a factor of 10 at epochs $x$ and $y$.

### A.5.1 Figure 1

We used E[BN] pruning in all models that were pruned, except for one model that used $\ell_2$-norm magnitude pruning, which was included in Figure 1 right but not displayed in Figure 1 left due to its qualitative similarity to prune$_S$ E[BN]. We leave out "E[BN]" in the legend of Figure 1 left, but all models nonetheless used E[BN] pruning.

The models were trained on CIFAR-10 with Adam for 325 epochs with $lr_s = (150, 300)$. The error bars are 95% confidence intervals for the mean, bootstrapped from 10 distinct runs of each experiment.

Since the layerwise pruning percentages varied, pruning required multiple iterative pruning percentages, the largest of which is denoted in the legend (rounded to the nearest integer).

**VGG11** Pruning targeted the final four convolutional layers during training with (layerwise) starting epochs $s = (3, 4, 5, 6)$, ending epochs $e = (150, 150, 150, 275)$, and pruning fractions $p = (0.3, 0.3, 0.3, 0.9)$. To allow for the same amount of pruning among models with differing iterative pruning percentages, we adjusted the number of inter-pruning retraining epochs. The models with the maximum iterative pruning percentage of 1% had $r = 4$, while the models with the maximum iterative pruning percentage of 13% had $r = 40$. The model pruned with $\ell_2$-norm magnitude pruning, which only appeared in Figure 1 right, had $r = 4$ as well.

**ResNet18** Pruning targeted the final four convolutional layers of ResNet18 during training with (layerwise) starting epochs $s = (3, 4, 5, 6)$, ending epochs $e = (150, 150, 170, 275)$, and pruning fractions $p = (0.25, 0.4, 0.25, 0.95)$. As noted in Section 4.1, we increased the pruning rate of the output layer of the penultimate block to remove shortcut connections to the last layer, thinking that it should increase the duration of adaptation to pruning. The models with the maximum iterative pruning percentage of 1% had $r = 4$, while the models with the maximum iterative pruning percentage of 13% had $r = 40$.

### A.5.2 FIGURE 2

Each experiment in Figure 2 targeted one of ten weight-magnitude deciles in the post-convolutional linear layer of the Conv4 network during training on CIFAR-10 with data augmentation.

While there are just ten deciles, the iterative nature of our pruning algorithms required the creation of eleven different pruning targets: ten methods pruned from the bottom of the decile upward (one experiment for each decile's starting point: 0th percentile, 10th percentile, etc.), and one (D10) pruned from the last decile's ending point downward (pruning the very largest collection of weights each iteration). In other words, D9 and D10 targeted the same decile (90th percentile to maximum value), but only D10 actually removed the largest weights on a given iteration (weights in the 100th-99th percentiles, for example). The D9 experiment would target weights starting from the 90th percentile (e.g. it may prune the 90th-91st percentiles on a particular iteration).

The training/pruning setup used the Adam optimizer, $s = (4)$, $e = (52)$, $p = (0.1)$, $r = 3$, and $lr_s = (30, 60)$. We calculated the generalization gap on epoch 54 and sampled average pruned magnitudes on epoch 35. We obtained qualitatively similar results regardless of whether we used fewer training epochs or data augmentation. The error bars are 95% confidence intervals for the means, bootstrapped from 10 distinct runs of each configuration.

### A.5.3 FIGURE 3

In Figure 3, prune$_L$ was applied to the final four convolutional layers of ResNet18 during training with (layerwise) starting epochs $s = (3, 4, 5, 6)$, ending epochs $e = (150, 150, 170, 275)$, and pruning fractions $p = (0.25, 0.4, 0.25, 0.95)$. Since the layerwise pruning percentages varied, pruning required multiple iterative pruning percentages, the largest of which is denoted in the legend (rounded to the nearest integer). The models with the maximum iterative pruning percentage of 1% had $r = 4$, the models with the maximum iterative pruning percentage of 13% had $r = 40$, and the "One Shot" model pruned all its targeted parameters at once on epoch 246.

When performing unstructured pruning, we pruned individual weights from filters based on their magnitude. The structured pruning experiments used E[BN] pruning.

The models were trained on CIFAR-10 with Adam for 325 epochs with $lr_s = (150, 300)$. The error bars are 95% confidence intervals for the means, bootstrapped from 10 distinct runs of each experiment.

### A.5.4 FIGURE 4

In Figure 4, pruning targeted the final four convolutional layers of VGG11 during training with (layerwise) starting epochs $s = (3, 4, 5, 6)$, ending epochs $e = (150, 150, 150, 275)$, and pruning fractions $p = (0.3, 0.3, 0.3, 0.9)$. To create the different iterative pruning rates, we used models with inter-pruning retrain periods $r = 4$, $r = 20$, $r = 40$, $r = 60$, and $r = 100$. Since the layerwise pruning percentages varied, pruning required multiple iterative pruning percentages, the largest of which is denoted on the horizontal axis. An unpruned baseline model average (10 runs) is plotted on the dotted line.

The models were trained on CIFAR-10 with Adam for 325 epochs with $lr_s = (150, 300)$. The error bars are 95% confidence intervals for the means, bootstrapped from 10 distinct runs of each experiment.

### A.5.5 FIGURE 5

In Figure 5, pruning targeted the final four convolutional layers of VGG11 during training with (layerwise) starting epochs $s = (3, 4, 5, 6)$, ending epochs $e = (150, 150, 150, 275)$, pruning fractions $p = (0.3, 0.3, 0.3, 0.9)$, and inter-pruning-iteration retrain period $r = 40$. When injecting pruning noise, we used the same pruning schedule and percentages, but applied noise to the parameters instead of removing them. The Gaussian noise had mean 0 and standard deviation equal to the empirical standard deviation of a noiseless filter from the same layer. Prune$_L$ used $\ell_2$-norm pruning.

The models were trained on CIFAR-10 with Adam for 325 epochs with $lr_s = (150, 300)$. The error bars are 95% confidence intervals for the means, bootstrapped from 10 distinct runs of each experiment.

### A.5.6 APPENDIX A.4

Each experiment in Appendix A.4 targeted the post-convolutional linear layer of the Conv4 network during training on CIFAR-10 with the Adam optimizer. The pruning algorithms start on epoch $s = (3)$, end on epoch $e = (18)$, prune the percentage $p = (0.9)$, and prune every epoch via retrain period $r = 1$. These relatively simple experiments were conducted to show that, at higher sparsity (pruning 85% of the model's parameters), unstable pruning can improve the generalization of the baseline. The error bars are 95% confidence intervals for the means, bootstrapped from 20 distinct runs of each configuration.

