# OpenReview forum: "The Generalization-Stability Tradeoff in Neural Network Pruning"
_ICLR.cc/2020/Conference — Reject_

### Official Review · AnonReviewer1 · 2019-10-21
**Official Blind Review #1**

**Rating:** 1

**Review:**

The paper is an empirical study that looks into the effect of neural network pruning on both the model accuracy as well as the generalization risk (defined as the difference between the training error and the test error). It concludes that while some pruning methods work, others fail. The authors argue that such discrepancy can be explained if we look into the impact of pruning on "stability".

The first major issue I have with the paper is in their definition of stability. I don't believe that this definition adds any value. Basically, the authors define stability by the difference between the test accuracy pre-pruning and post-pruning. This makes the results nearly tautological (not very much different from claiming that the test accuracy changes if the test accuracy is going to change!). One part where this issue is particularly important is when the authors conclude that "instability" leads to an improved performance. However, if we combine both the definition of test accuracy and the definition of "instability" used the paper, what the authors basically say is that pruning improves performance. To see this, note that a large instability is equivalent to the statement that the test accuracy changes in any direction (there is an absolute sign). So, the authors are saying that the test accuracy after pruning improves if it changes, which is another way of saying that pruning helps.

The second major issue is that some of the stated contributions in the paper are not discussed in the main body of the paper, but rather in the appendix. For example, the authors mention that one of their contributions is a new pruning method but that method is not described in the paper at all, only in the appendix. If it is a contribution, the authors should include it in the main body of the paper.

Third, there are major statements in the paper that are not well-founded. Take, for example, the experiment in Section 4.4, where they apply zeroing noise multiple times. The authors claim that since the weights are only forced to zero every few epochs, the network should have the same capacity as the full network (i.e. VC capacity). I disagree with this. The capacity should reduce since those weights are not allowed to be optimized and they keep getting reset to zero every few epochs. They are effectively as if they were removed permanently.



**Experience Assessment:**

I have published one or two papers in this area.

**Review Assessment: Checking Correctness Of Derivations And Theory:**

N/A

**Review Assessment: Checking Correctness Of Experiments:**

I assessed the sensibility of the experiments.

**Review Assessment: Thoroughness In Paper Reading:**

N/A

---

> ### Author Response · Authors · 2019-11-12
> **Response to Reviewer 1**
>
> We thank the reviewer for their helpful comments.
>
> **The Absence of a Tautology**
>
> To better convey our findings, we would like to clarify how we measure test accuracy and instability. The generalization-stability tradeoff is essentially a positive correlation between test accuracy at convergence (i.e., generalization), and the average drop in test accuracy from immediately before to immediately after pruning (i.e., average instability, where the average is computed across all pruning events). Put another way, we are *not* investigating the relationship between an increase in test accuracy following a pruning event and the final increase in test accuracy. Rather, we found that when pruning immediately causes large *drops* in test accuracy (high instability), converged (or final) test accuracy increases. To make this more apparent in our main text, we removed the absolute value component of the instability calculation. Additionally, we added to Section 3 a timeline that shows when instability is calculated with respect to pruning events.
>
> Removing the absolute value component of the instability calculation had no qualitative effect on the demonstrated relationship between instability and final generalization (in Figure 1, the statistical significance of the correlation actually increased). Note that our unchanged results are unsurprising, as negative instabilities were rare to begin with; i.e., a negative instability implies that pruning somehow had an immediately positive effect on test accuracy, which should be (intuitively) and is (see the updated Figure A2) rare. In any case, we believe that refraining from taking the absolute value has made the suggested correlation between accuracy drops and generalization (highlighted with arrows in Figure 1’s accuracy plots) more clearly supported by the statistical correlations (shown in Figure 1’s correlation plots).
>
> We hope to have made clear that our main finding lacks a tautological nature. Indeed, that higher final test accuracy (generalization) is facilitated by pruning that causes larger immediate damage to the test accuracy (instability) is not at all tautological (or obvious). Correspondingly, when used in practice for model compression, magnitude pruning almost always targets small weights [1, 2, 5, 6]. Similarly, we are unaware of any suggestion that DNN pruning instability might be helpful—-the canonical approaches OBD and OBS actually use second-order information about the loss to ensure pruning is as *stable* as possible [3, 4]. Please let us know if you are aware of a DNN pruning paper that advocates any disruptiveness/instability in pruning, as we would like to add it to our related work.
>
> **Moving Derivation from Appendix**
>
> While we discussed all of our contributions in the main text, we chose to put the mathematical derivation of our novel pruning target in the appendix to save space. However, we agree with the reviewer that this derivation is critical to the contribution, and have therefore moved the derivation into the main text. Please note that the paper is a little longer than 8 pages as a result of this change.
>
> **The Capacity Effect of Temporarily Noising/Removing Weights**
>
> We wish to make clear that the pruning noise we apply affects a particular weight for no more than one segment of training time, where a segment is a series of N consecutive batches (we edited our main text to make this more clear). After a weight has been zeroed/noised for N batches, it resumes normal training and can contribute to the model again. That said, the reviewer is correct in that weights that were temporarily zeroed do not necessarily learn upon reentering the model, so our approach may effectively remove some weights. However, we observed (see Figure 5) that pruning the reentered weights at convergence resulted in a marked drop in performance (for all noise schemes except “Zeroing 1500”), showing that the reentered weights had typically learned after reentry, and that temporary zeroing is therefore less harmful to capacity than permanent pruning. We added this explanation to our main text.

---

> > ### Author Response · Authors · 2019-11-12
> > **References**
> >
> > [1] Jonathan Frankle and Michael Carbin. The lottery ticket hypothesis: Finding sparse, trainable
> > neural networks. arXiv preprint arXiv:1803.03635, 2018.
> > [2] Song Han, Jeff Pool, John Tran, and William Dally.  Learning both weights and connections
> > for efficient neural network. In Advances in neural information processing systems, pages 1135–1143, 2015.
> > [3] Babak Hassibi and David G Stork. Second order derivatives for network pruning: Optimal brain surgeon. In Advances in neural information processing systems, pages 164–171, 1993.
> > [4] Yann LeCun, John S Denker, and Sara A Solla. Optimal brain damage. In Advances in neural
> > information processing systems, pages 598–605, 1990.
> > [5] Hao Li, Asim Kadav, Igor Durdanovic, Hanan Samet, and Hans Peter Graf. Pruning filters for
> > efficient convnets. arXiv preprint arXiv:1608.08710, 2016.
> > [6] Michael Zhu and Suyog Gupta. To prune, or not to prune: exploring the efficacy of pruning for
> > model compression. arXiv preprint arXiv:1710.01878, 2017.

---

### Official Review · AnonReviewer3 · 2019-10-22
**Official Blind Review #3**

**Rating:** 3

**Review:**

This paper mainly studies the relationship between the generalization error and mean/variance of the test accuracy. The authors first propose a new score for pruning called E[BN]. Then, the authors observe the generalization error and the test accuracy mean/variance for pruning large score weights and small score weights for VGG11, ResNet18, and Conv4 models. From these experiments, the authors observe that pruning large score weights generates instable but high test accuracy and smaller generalization gap compared to pruning small score weights. The authors additionally study some other aspects of pruning (e.g., pruning as a noise injection) and conclude the paper.

Overall, I am not sure whether the observation holds in general due to the below reasons.

- The authors proposed a new score E[BN] and all experiments are performed on this. However, how it differs from the usual magnitude-based pruning in practice is unclear. I would like to know whether similar behavior is observed for the naïve magnitude-based pruning.

- I think that the author’s observation is quite restricted and cannot extend to a general statement since the experiments are only done for pruning small score/large score weights. To verify the generalization and instability trade-off, I believe that it is necessary to examine several (artificial) pruning methods controlling the instability of test accuracies and check whether the proposed trade-off holds. For example, one can design pruning methods that disconnect (or almost disconnect) the network connection from the bottom to the top (i.e., pruned network always outputs constant) with some probability to extremely increase the instability.

- The authors did not report the results for high sparsity.

Besides, I am not sure the meaning of the instability since when the test accuracy of the pruned model is higher than that of the unpruned model, the instability could be large.

Other comments:
- The first paragraph mentions that the generalization gap might be a function of the number of parameters. However, I think that it is quite trivial that the generalization gap is not a function of the number of parameters while it only provides the upper bound.
----------------------------------------------------------
I have read the authors' response. Thanks for clarifying the definition of instability and additional experiments with high sparsity. However, I will maintain my score due to the following concern.

The remaining concern is that the current evidence for verifying generalization-stability tradeoff is not convincing as the authors presented only some examples having small and large instability (e.g., pruning smallest/largest weights) under the same pruning algorithm. I think that the results would be more convincing if the authors add a test accuracy plots given a fixed prune ratio, whose x-axis is controlled instabilities (e.g., from 10% to 90%) among various pruning algorithms (other than magnitude-based ones, e.g., Hessian based methods). It would be much more interesting if the same instability results same test accuracy even for different pruning algorithms.


**Experience Assessment:**

I have read many papers in this area.

**Review Assessment: Checking Correctness Of Derivations And Theory:**

N/A

**Review Assessment: Checking Correctness Of Experiments:**

I assessed the sensibility of the experiments.

**Review Assessment: Thoroughness In Paper Reading:**

I read the paper at least twice and used my best judgement in assessing the paper.

---

> ### Author Response · Authors · 2019-11-12
> **Response to Reviewer 3**
>
> We thank the reviewer for their helpful comments.
>
> **The Use of E[BN] vs. Naive Approaches**
>
> While Figure 1 used the proposed E[BN] score, Figures 2, 3, and 4 contain results from experiments with L2-norm pruning and simple-magnitude pruning. For example, the two plots in Figure 4 that are entitled “L2 Norm Pruning” contain results from L2-norm pruning experiments. To illustrate the generalization-stability tradeoff across pruning methods more thoroughly, we added to the updated paper a version of Figure 1 that uses L2-norm pruning (see Figure A3). Figure A3 has qualitatively similar results to Figure 1, but the correlation between instability and generalization is somewhat weaker in Figure A3. This may be explained by the fact that L2-norm pruning generated a narrower spectrum of instabilities (perhaps due to L2-norm scoring’s inability to accurately assess parameter importance, which was shown in Figure 4).
>
> **The Scope of Our Experiments, and Adding Higher Sparsity Results**
>
> We would like to clarify that our conclusions are *not* based only on comparisons of large/small weight pruning, but rather that we used dozens of experimental configurations to (across three model classes) compare: 1) pruning of small score, random score, and large score weights (Figure 4; also, Figure 2 finely differentiated scores by pruning specific score deciles); 2) various iterative pruning rates, and therefore pruning schedules (Figure 4); 3) structured pruning vs. unstructured pruning in two different contexts—-unstructured vs. structured ResNet filter pruning (Figure 3), and unstructured vs. structured Conv4 linear-layer pruning (Figure 2); 4) L2-norm vs. E[BN] weight scoring (Figure 4); and 5) iterative pruning vs. one-shot pruning (Figure 3). While all of our experiments used some form of magnitude pruning, this form of pruning is perhaps the most common [2, 4–6], and has been shown to perform similarly to more sophisticated pruning approaches [3] (as mentioned in our related work section). Thus, our experiments observed the generalization-stability relationship across a large range of interesting contexts and instability values (see Figures A2 and 4, for examples of such values). Importantly though, as we stated in our discussion section, adding instability can have downsides, so the best generalizing pruning approach may involve some mixture of unstable and stable pruning.
>
> To evaluate our results under higher sparsity, we used high- and low-stability pruning algorithms to prune 85% of Conv4, and we found that the generalization-stability tradeoff was present. We've added this result in Appendix A4.
>
> **The Meaning of Instability**
>
> We wish to note that instability does not take into account final test accuracy. To make clear that instability is calculated from test accuracies computed immediately before and after pruning, we added to Section 3 a timeline that depicts when instability is calculated relative to pruning iterations.
>
> To add additional clarity, we removed the absolute value component of the instability calculation. Thus, it's no longer possible for instability to be positive when pruning immediately helps the model. Importantly, the updated version of Figure A2 shows that instability is rarely negative (i.e., it’s rare that pruning immediately helps the model), and never negative and large. Perhaps unsurprisingly, then, removing the absolute value did not notably affect our conclusions, in fact doing so improved the statistical evidence for the tradeoff (see Figure 1).
>
> **The Role of the Generalization Gap and Bounds**
>
> While generalization gap bounds are loose and imprecise (they are just bounds, as the reviewer points out), they have nonetheless been used to guide model selection (see the discussion of this in our related work section). We updated the introduction to make more clear our point that some recent empirical and theoretical results (e.g. generalization gap bounds) now similarly guide us to select models that are overparameterized. This new guidance seems at odds with more traditional ideas (such as the idea that pruning improves generalization via removing parameters). For a discussion of the conflict between traditional and recent recommendations on choosing a model size (or parameter count), see [1]: “a best practice in deep learning for choosing neural network architectures [is] that the network should be large enough to permit effortless zero loss training… in direct challenge to the bias-variance trade-off philosophy...”
>
> We also wish to note that, except in Figure 2, we exclusively look at generalization to a test set (defined as the test accuracy), not the generalization gap.

---

> > ### Author Response · Authors · 2019-11-12
> > **References**
> >
> > [1] Mikhail Belkin, Daniel Hsu, Siyuan Ma, and Soumik Mandal.  Reconciling modern machine
> > learning and the bias-variance trade-off. arXiv preprint arXiv:1812.11118, 2018.
> > [2] Jonathan Frankle and Michael Carbin. The lottery ticket hypothesis: Finding sparse, trainable
> > neural networks. arXiv preprint arXiv:1803.03635, 2018.
> > [3] Trevor Gale, Erich Elsen, and Sara Hooker. The state of sparsity in deep neural networks.
> > CoRR, abs/1902.09574, 2019. URL http://arxiv.org/abs/1902.09574.
> > [4] Song Han, Jeff Pool, John Tran, and William Dally.  Learning both weights and connections
> > for efficient neural network.   In Advances in neural information processing systems,  pages 1135–1143, 2015.
> > [5] Hao Li, Asim Kadav, Igor Durdanovic, Hanan Samet, and Hans Peter Graf. Pruning filters for
> > efficient convnets. arXiv preprint arXiv:1608.08710, 2016.
> > [6] Michael Zhu and Suyog Gupta. To prune, or not to prune: exploring the efficacy of pruning for
> > model compression. arXiv preprint arXiv:1710.01878, 2017.

---

### Official Review · AnonReviewer2 · 2019-10-23
**Official Blind Review #2**

**Rating:** 1

**Review:**

This paper studies a puzzling question: if larger parameter counts (over-parameterization) leads to better generalization (less overfitting), how does pruning parameters improve generalization? To answer this question, the authors analyzed the behaviour of pruning over training and finally attribute the pruning's effect on generalization to the instability it introduces.

I tend to vote for a rejection because
(1) The explanation of instability and noise injection is not new. Pruning algorithms have long been interpreted from Bayesian perspective. Some parameters with large magnitude or large importance contribute to large KL divergence with the prior (or equivalently large description length), therefore it's not surprising that removing those weights would improve generalization.
(2) To my knowledge, the reason why over-parameterization improves generalization (or reduces overfitting) is because over-parameterized networks can find good solution which is close to the initialization (the distance to the initialization here can be thought of as a complexity measure). In this sense, the effect of over-parameterization is on neural network training. However, pruning is typically conducted after training, so I don't think the fact that pruning parameters improves generalization contradicts the recent generalization theory of over-parameterized networks. Particularly, these two phenomena can both be explained from Bayesian perspective.

**Experience Assessment:**

I have published in this field for several years.

**Review Assessment: Checking Correctness Of Derivations And Theory:**

N/A

**Review Assessment: Checking Correctness Of Experiments:**

I assessed the sensibility of the experiments.

**Review Assessment: Thoroughness In Paper Reading:**

I read the paper at least twice and used my best judgement in assessing the paper.

---

> ### Author Response · Authors · 2019-11-12
> **Response to Reviewer 2**
>
> We thank the reviewer for their helpful comments.
>
> **Bayesian Pruning Studies’ Relationship to Our Study and the Generalization-Stability Tradeoff**
>
> The reviewer notes that the effect of noise on generalization has been studied. We wish to make clear that our related work and discussion sections identified the novel role of noise in our analysis, which required making comparisons to studies of dropout, noise injection, and pruning [2, 3, 5, 6, 10, 11]. In particular, in the last paragraph of our related work section, we stated that the noise in Bayesian/dropout pruning “zeroes” and restores (or otherwise noises) the weights’ values very frequently (e.g., a new weight vector is sampled for every single data point in a batch [2, 5, 6]), *but* (among other differences) the noise in *iterative* pruning permanently zeroes subsets of the weights and does so comparatively rarely (e.g., a new subset of the weight vector is replaced with zeroes at roughly ten distinct points during training). To make our study’s focus more clear, we changed the phrase “pruning noise” in this section to “iterative DNN pruning noise”. Please let us know if you are aware of any papers that frame *iterative* DNN pruning as noise injection, as we believe ours is the first. Further, we believe this contribution is important because iterative pruning is, to the best of our knowledge, the most popular DNN pruning approach in the literature and deployment (e.g., TensorFlow’s pruning library uses iterative magnitude pruning [14]).
>
> The reviewer notes that the KL-divergence mechanism from the Bayesian literature makes our main finding (the presence of a generalization-stability tradeoff) unsurprising. Our understanding, however, is that larger magnitude weights need not cause larger KL divergences. For example, Bayesian pruning with a log-uniform prior has the effect of “essentially either pruning the parameter or keeping it close to the maximum likelihood estimate” [5]. Furthermore, even with a prior that discourages large weights, shrinking the cost of communicating the model (the KL divergence between the weights and their prior) is balanced with shrinking the cost of communicating prediction errors [5]. Thus, large weights can survive Bayesian pruning and weight shrinkage mechanisms if these weights are important to accuracy, suggesting Bayesian pruning is quite different than the unstable pruning we analyze, which is (at least initially) quite damaging to the accuracy. Notably, in the Bayesian literature, pruning that leaves large weights intact is also empirically supported: [5] found that the horseshoe prior, which encourages unpruned weights to be near 0 and “can, potentially, offer better regularization and generalization”, actually led to worse generalization than the log-uniform prior, which doesn't aim to shrink large weights. Similarly, [6] discussed sparsity, not magnitude shrinkage, as the source of the generalization benefits in their Bayesian pruning approach.
>
> Importantly, while all of the pruning studies we’re aware of avoid pruning large weights in order to minimize harm to the loss, our study has shown that harm to the loss may be related to how pruning engenders better generalization. As such, our conclusion suggested that pruning algorithms may be improved by balancing the apparent generalization benefits of short-term accuracy degradation with the risks such unstable pruning entails (e.g., damaging capacity too much). If the reviewer is aware of a study (Bayesian or not) showing unstable pruning improving generalization, or a study recommending the pruning of the most important weights, then please let us know of the relevant paper(s), as we would like to add such studies to our related work.
>
> **Overparameterization and Pruning**
>
> We wish to clarify that, since we are primarily considering iterative pruning, the model becomes smaller as training progresses, making it unclear whether the benefit to training of overparameterization (that was identified by the reviewer) is diminished by the pruning process. This motivates the puzzling question we posed, and we made this clearer by updating the introduction to reflect that we are talking about pruning throughout training. Also, while the reviewer’s understanding of how overparameterization helps generalization is valuable (and consistent with some of the work we cited; for example [7]), our understanding is that this is an active area of research: “Despite existing work on ensuring generalization of neural networks in terms of scale sensitive complexity measures, such as norms, margin and sharpness, these complexity measures do not offer an explanation of why neural networks generalize better with overparameterization” [9]; also, see [1, 4, 8, 12, 13].

---

> > ### Author Response · Authors · 2019-11-12
> > **References**
> >
> >
> > [1] Mikhail Belkin, Daniel Hsu, Siyuan Ma, and Soumik Mandal. Reconciling modern machine
> > learning and the bias-variance trade-off. arXiv preprint arXiv:1812.11118, 2018.
> > [2] Aidan N Gomez, Ivan Zhang, Kevin Swersky, Yarin Gal, and Geoffrey E Hinton.  Targeted
> > dropout. 2018.
> > [3] Geoffrey E Hinton and Drew Van Camp. Keeping the neural networks simple by minimizing
> > the  description  length  of  the  weights.   In Proceedings of the sixth annual conference on
> > Computational learning theory, pages 5–13. ACM, 1993.
> > [4] Nitish Shirish Keskar, Dheevatsa Mudigere, Jorge Nocedal, Mikhail Smelyanskiy, and Ping
> > Tak Peter Tang. On large-batch training for deep learning: Generalization gap and sharp minima.
> > arXiv preprint arXiv:1609.04836, 2016.
> > [5] Christos Louizos, Karen Ullrich, and Max Welling. Bayesian compression for deep learning. In Advances in Neural Information Processing Systems, pages 3290–3300, 2017.
> > [6] Dmitry Molchanov, Arsenii Ashukha, and Dmitry Vetrov. Variational dropout sparsifies deep
> > neural networks. arXiv preprint arXiv:1701.05369, 2017.
> > [7] Vaishnavh Nagarajan and J Zico Kolter. Generalization in deep networks: The role of distance
> > from initialization. arXiv preprint arXiv:1901.01672, 2019.
> > [8] Behnam Neyshabur, Ryota Tomioka, and Nathan Srebro. In search of the real inductive bias:
> > On the role of implicit regularization in deep learning. arXiv preprint arXiv:1412.6614, 2014.
> > [9] Behnam  Neyshabur,  Zhiyuan  Li,  Srinadh  Bhojanapalli,  Yann  LeCun,  and  Nathan  Srebro. Towards understanding the role of over-parametrization in generalization of neural networks. arXiv preprint arXiv:1805.12076, 2018.
> > [10] Ben Poole, Jascha Sohl-Dickstein, and Surya Ganguli. Analyzing noise in autoencoders and
> > deep networks. arXiv preprint arXiv:1406.1831, 2014.
> > [11] Nitish Srivastava, Geoffrey Hinton, Alex Krizhevsky, Ilya Sutskever, and Ruslan Salakhutdinov. Dropout: A simple way to prevent neural networks from overfitting. The Journal of Machine Learning Research, 15(1):1929–1958, 2014.
> > [12] Zhewei Yao, Amir Gholami, Qi Lei, Kurt Keutzer, and Michael W Mahoney. Hessian-based
> > analysis of large batch training and robustness to adversaries. In Advances in Neural Information Processing Systems, pages 4949–4959, 2018.
> > [13] Chiyuan Zhang, Samy Bengio, Moritz Hardt, Benjamin Recht, and Oriol Vinyals. Understanding deep learning requires rethinking generalization. arXiv preprint arXiv:1611.03530, 2016.
> > [14] Michael Zhu and Suyog Gupta. To prune, or not to prune: exploring the efficacy of pruning for model compression. arXiv preprint arXiv:1710.01878, 2017.

---

> ### Author Response · Authors · 2019-11-14
> **Response to Reviewer 2, Addendum**
>
> Again, we thank the reviewer for their helpful comments.
>
> While Section 5 of our original submission included mention of the Bayesian mechanism suggested by the reviewer (see our reference to [1]), we have made the potential role of this mechanism more prominent by connecting it to our results in Section 4.3. Illuminating the particular mechanism giving rise to the generalization-stability tradeoff is (as our discussion stated) important, but it's perhaps secondary in importance to simply identifying the tradeoff because, when used in practice for model compression, magnitude pruning almost always targets small weights [2, 3, 4, 5]. While we agree with the reviewer on the potential applicability of the Bayesian mechanism to our results, then, we maintain our original disagreement with the suggestion that usage of Bayesian priors in the literature makes unsurprising our finding that large weight pruning is helpful.
>
> [1] Geoffrey E Hinton and Drew Van Camp. Keeping the neural networks simple by minimizing the description length of the weights. In Proceedings of the sixth annual conference on Computational learning theory, pages 5–13. ACM, 1993.
> [2] Song Han, Jeff Pool, John Tran, and William Dally. Learning both weights and connections for efficient neural network. In Advances in neural information processing systems, pages 1135–1143, 2015.
> [3] Hao Li, Asim Kadav, Igor Durdanovic, Hanan Samet, and Hans Peter Graf. Pruning filters for efficient convnets. arXiv preprint arXiv:1608.08710, 2016.
> [4] Michael Zhu and Suyog Gupta. To prune, or not to prune: exploring the efficacy of pruning for model compression. arXiv preprint arXiv:1710.01878, 2017.
> [5] Jonathan Frankle and Michael Carbin. The lottery ticket hypothesis: Finding sparse, trainable neural networks. arXiv preprint arXiv:1803.03635, 2018.

---

### Decision · Program_Chairs · 2019-12-19

**Decision:**

Reject

**Comment:**

The authors introduce a notion of stability to pruning and argue through empirical evaluation that pruning leads to improved generalization when it introduces instability. The reviewers were largely unconvinced, though for very different reasons. The idea that "Bayesian ideas" explain what's going on seems obviously wrong to me. The third reviewer seems to think there's a tautology lurking here and that doesn't seem to be true to me either. It is disappointing that the reviewers did not re-engage with the authors after the authors produced extensive rebuttals. Unfortunately, this is a widespread pattern this year.

Even though I'm inclined to ignore aspects of these reviews, I feel that there needs to be a broader empirical study to confirm these findings. In the next iteration of the paper, I believe it may also be important to relate these ideas to [1]. It would be interesting to compare also on the networks studied in [1], which are more diverse.


[1] The Lottery Ticket Hypothesis at Scale (Jonathan Frankle, Gintare Karolina Dziugaite, Daniel M. Roy, and Michael Carbin) https://arxiv.org/abs/1903.01611